# Net decrease in spine-surface GluA1-containing AMPA receptors after post-learning sleep in the adult mouse cortex

Daisuke Miyamoto [1,3], William Marshall[1,2], Giulio Tononi [1✉] & Chiara Cirelli [1✉]

The mechanisms by which sleep benefits learning and memory remain unclear. Sleep may further strengthen the synapses potentiated by learning or promote broad synaptic weakening while protecting the newly potentiated synapses. We tested these ideas by combining a motor task whose consolidation is sleep-dependent, a marker of synaptic AMPA receptor plasticity, and repeated two-photon imaging to track hundreds of spines in vivo with single spine resolution. In mouse motor cortex, sleep leads to an overall net decrease in spine-surface GluA1-containing AMPA receptors, both before and after learning. Molecular changes in single spines during post-learning sleep are correlated with changes in performance after sleep. The spines in which learning leads to the largest increase in GluA1 expression have a relative advantage after post-learning sleep compared to sleep deprivation, because sleep weakens all remaining spines. These results are obtained in adult mice, showing that sleep-dependent synaptic down-selection also benefits the mature brain.

[1] Department of Psychiatry, University of Wisconsin-Madison, Madison, WI, USA. [2] Department of Mathematics and Statistics, Brock University, St. Catharines, ON, Canada. [3] Present address: University of Toyama, Toyama, Japan. ✉email: gtononi@wisc.edu; ccirelli@wisc.edu

There is ample evidence that sleep benefits cognitive functions. Lack of sleep impairs many aspects of cognition, from response time, working memory, and sustained attention, to executive functions, affect, and mood[1–4]. On the other hand, sleep promotes memory consolidation and integration, gist extraction as well as forgetting, and restores the ability to learn[5,6]. Still, the synaptic mechanisms underlying the positive effects of sleep on cognition are poorly characterized[7–9]. Two main hypotheses have been proposed. One is that sleep consolidates memories by further strengthening the synaptic connections potentiated by learning[5,7]. According to this view, sleep-dependent potentiation is mediated by the sequential reactivation of specific sets of neurons and synapses, which occurs during hippocampal sharp-wave ripples and during the UP states of the cortical slow oscillations[10–12]. The other hypothesis is that sleep promotes broad synaptic weakening while still protecting from renormalization the synapses coherently reactivated during sleep, including those engaged by new learning[6,13,14]. This down-selection process results in the relative strengthening of recently activated synapses, without the need for an absolute increase in their efficacy. By avoiding synaptic saturation, the same down-selection process can also explain why sleep restores the capacity for new learning, which cannot be accounted for by sleep-dependent potentiation alone.

The majority of synapses in the mammalian brain are excitatory, established by glutamatergic axons that contact protrusions of the dendritic shaft called spines[15,16]. Whether the weakening of some axospinous synapses during sleep can co-occur with the absolute strengthening of other synapses remains unclear. The few available studies of well-characterized brain regions, including primary motor cortex, measured different aspects of synapses, such as number or size, under different waking conditions and learning paradigms, after different periods of wake and sleep, and at different developmental ages. Specifically, one study in adolescent mice found that, after motor training during the day, spinogenesis was impaired by a few hours of sleep deprivation, but continued at similar levels whether motor training was followed predominantly by sleep (during the day) or by wakefulness (during the night)[17]. The study focused on the apical tuft dendrites of layer 5 pyramidal neurons and did not measure changes in spine size, which are substantial even in the adult cortex, outnumber the changes in spine number[18,19], and are correlated with changes in synaptic strength[15,20]. On the other hand, two other studies of mouse primary motor cortex[21,22] found a sleep-dependent decrease in different markers of synaptic strength, but did not assess the effects of sleep after a specific learning task, nor did they follow the same synapse after defined periods of waking and sleep.

In the present study, we aimed at characterizing the effects of sleep on memory consolidation by tracking longitudinally many spines with single spine resolution, both during the sleep-wake cycle and in response to learning. To do so, we applied repeated two-photon imaging in vivo to measure spine number and size, as well as the expression of a molecular marker of synaptic strength, in layer 2/3 of the mouse primary motor cortex after several hours of sleep without prior motor training, immediately after motor learning, and after several hours of post-learning sleep or sleep deprivation. We targeted the superficial layers because they are known to remain highly plastic even after early development (e.g., refs. [23–27]). We find that sleep still leads to a net decrease in synaptic strength even after skill learning, and the weakening of most spines, those not directly engaged by learning, accounts for sleep-dependent offline consolidation.

## Results

Learning and the induction of long-term potentiation are associated with a sustained increase in the number and activity of the glutamatergic AMPA receptors. A key step in this process is the incorporation of GluA1-containing AMPA receptors in the post-synaptic membrane, often in association with the growth of the dendritic spine harboring the synapse[28,29]. We used in utero electroporation of mouse embryos (aged 14.5 embryonic days) to transfect layer 2/3 pyramidal neurons of primary motor cortex with the red fluorescent protein dsRed2, a morphological marker that labels dendritic arbors and spines, and with the GluA1 subunit of the AMPA receptors tagged with Super Ecliptic pHluorin (SEP[30]). SEP is a pH-sensitive variant of the green fluorescent protein that reports cell-surface receptors selectively[28,29]. Low concentrations of DNA were used for electroporation to minimize the overexpression of AMPA receptors, and GluA2 was included to approximate the natural GluA1/GluA2 ratio. Immunostaining of GluA1 in slices of electroporated mice confirmed that GluA1 was only minimally overexpressed (Supplementary Fig. 1). Transfected mice were implanted with a cranial window over the primary motor cortex at around two months of age (postnatal day 56–78) and allowed to recover for 2–3 weeks before repeated two-photon imaging started (Fig. 1a). Mice (n = 12) were imaged several times in the course of two consecutive days (Fig. 1a), starting at the beginning of the light phase, 7 h later, and the following morning immediately after training in a complex wheel task that engages the primary motor cortex[31]. After training mice were split in two groups (n = 6 mice/group) that were either kept awake or allowed to sleep for 6–7 h. Afterward all animals were imaged again and then left undisturbed until a second training session occurred the next morning, 24 h after the first session, followed by the final imaging session.

**Sleep promotes the consolidation of motor learning.** Sleep/waking behavior was constantly monitored across the 48 h. During the first day all mice were mostly asleep during the first 6–7 h of the light phase (−24 h to −17 h) and awake at night (−12 h to 0 h), as expected for nocturnal animals entrained to the 12:12 light/dark cycle (Fig. 1b, c). After motor training the next morning, mice left undisturbed were again mostly asleep for 6–7 h, while during the same post-training period sleep deprived mice were awake 99% of the time (0 h to 7 h; Fig. 1b, c). In the motor task animals trained to run on a wheel equipped with an uneven rung pattern that rotates at increasing speed[31]. The first training session lasted ~1 h, and performance in all mice improved by the end of the session (first 3 trials vs. last 3 trials, paired t test; p = 0.0004, n = 12 mice; Fig. 1d; Supplementary Table 1), with no difference in the mean performance across all trials between the two groups (Student's t-test, S vs. SD; p = 0.4977). The next day, by contrast, there was a significant difference between the S and SD groups in the mean performance across the first 3 trials of the second session (Student's t test, S vs. SD; p = 0.0236), with the sleeping mice outperforming the sleep deprived mice (Fig. 1e; Supplementary Table 1). Thus, as in our previous study[31], we found that sleep promotes the consolidation of the complex wheel task.

**The balance between newly formed and lost spines is unaffected by sleep or learning.** The first imaging session, which occurred at light onset (−24 h), confirmed that only a small number of pyramidal neurons in layer 2/3 were transfected, leading to sparse labeling of dsRed2 and SEP-GluA1 in the dendritic shafts (Fig. 1a). Moreover, spine size (dsRed2 expression) and SEP-GluA1 expression varied greatly among spines and followed a lognormal distribution (12 mice, 1530 spines; Fig. 2a; Supplementary Fig. 2), in agreement with previous data in mouse cortex[21]. DsRed2 and SEP-GluA1 expression were strongly positively correlated (Fig. 2b; Supplementary Table 2), consistent

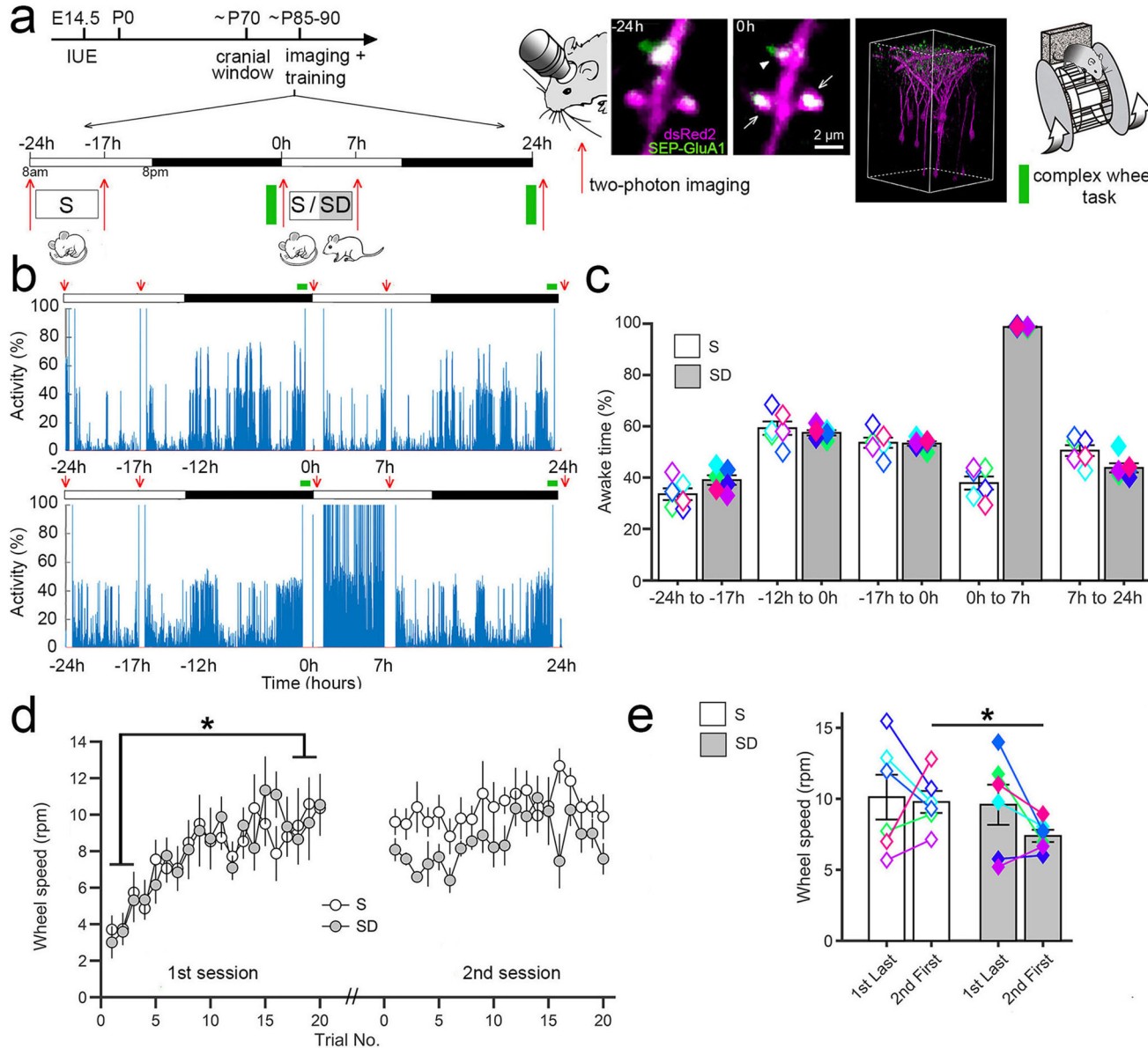

**Fig. 1 Sleep/waking behavior and motor learning. a** Left, experimental design. White and black bars indicate the light and dark periods, respectively. IUE, in utero electroporation. E and P indicate embryonic and postnatal age, respectively, in days. S, sleep; SD, sleep deprivation. Right, representative examples of time-lapse in vivo two-photon images of layer 2/3 pyramidal cell apical dendrites, representative 3D reconstruction showing sparse labeling in layer 2/3 ($X$, $Y = 200 \, \mu m$, $Z = 300 \, \mu m$; 12 mice in total), and schematic of the complex wheel used for motor training. SEP-GluA1 (green), dsRed2 (red), overlap (white). **b** Representative examples of rest/activity patterns in one S and one SD mouse during the 2 consecutive days when repeated two-photon imaging and motor training occurred (red arrows). **c** Time spent awake (% of total time; mean ± SEM; 6 S, 6 SD mice) during the indicated time intervals (−24 h to −17 h = 36.3 ± 1.7; −12 h to −0 h = 61.7 ± 1.3; −17 h to −0 h = 56.8 ± 1.0; 0 h to 7 h, S mice = 37.9 ± 2.8; SD mice = 98.7 ± 0.2; 7 h to 24 h, S mice = 50.5 ± 2.3; SD mice = 43.8 ± 2.0). Colored symbols indicate individual animals. **d** Performance in the complex wheel task for each trial, averaged (± SEM) across the 6 mice of each group. Performance in the first session (first 3 trials vs. last 3 trials): *, two-sided paired $t$ test, $p = 0.0004$ ($n = 12$ mice). **e** Offline consolidation measured by comparing the first 3 trials of session 2 between S and SD mice (mean ± SEM); *, two-sided Student's $t$ test, $p = 0.0236$. Colored symbols indicate individual animals. Source data are provided as a Source data file.

with the close link between spine size and synaptic strength; a weaker, but significant correlation was also present between SEP-GluA1 intensity in spines and in the adjacent shaft (Fig. 2b; Supplementary Table 2).

We first counted the number of spines newly formed or eliminated in each imaging session relative to the previous one, and found that both spine formation and spine elimination were rare and balanced at all times (formation vs. elimination, Wilcoxon signed-rank test; $p > 0.3$ for all time points in both S and SD mice) (Fig. 2c). Across the entire experimental period

(48 h), persistent spines accounted for 98.8 ± 0.60% of all spines in S mice and for 98.7 ± 0.60% in SD mice (Fig. 2c).

**Sleep leads to a net decrease in SEP-GluA1 expression.** Next, we measured the normalized difference (ND = ($x_{post} - x_{pre}$)/($x_{post} + x_{pre}$)) in SEP-GluA1 expression in each spine before and after 7 h of sleep in the absence of any motor training (pre-learning sleep, −24 h to −17 h). We applied a linear mixed effect (LME) model that used time as a categorical fixed effect, and spine, dendrite, and mouse as random effects (see "Methods"). Across all spines,

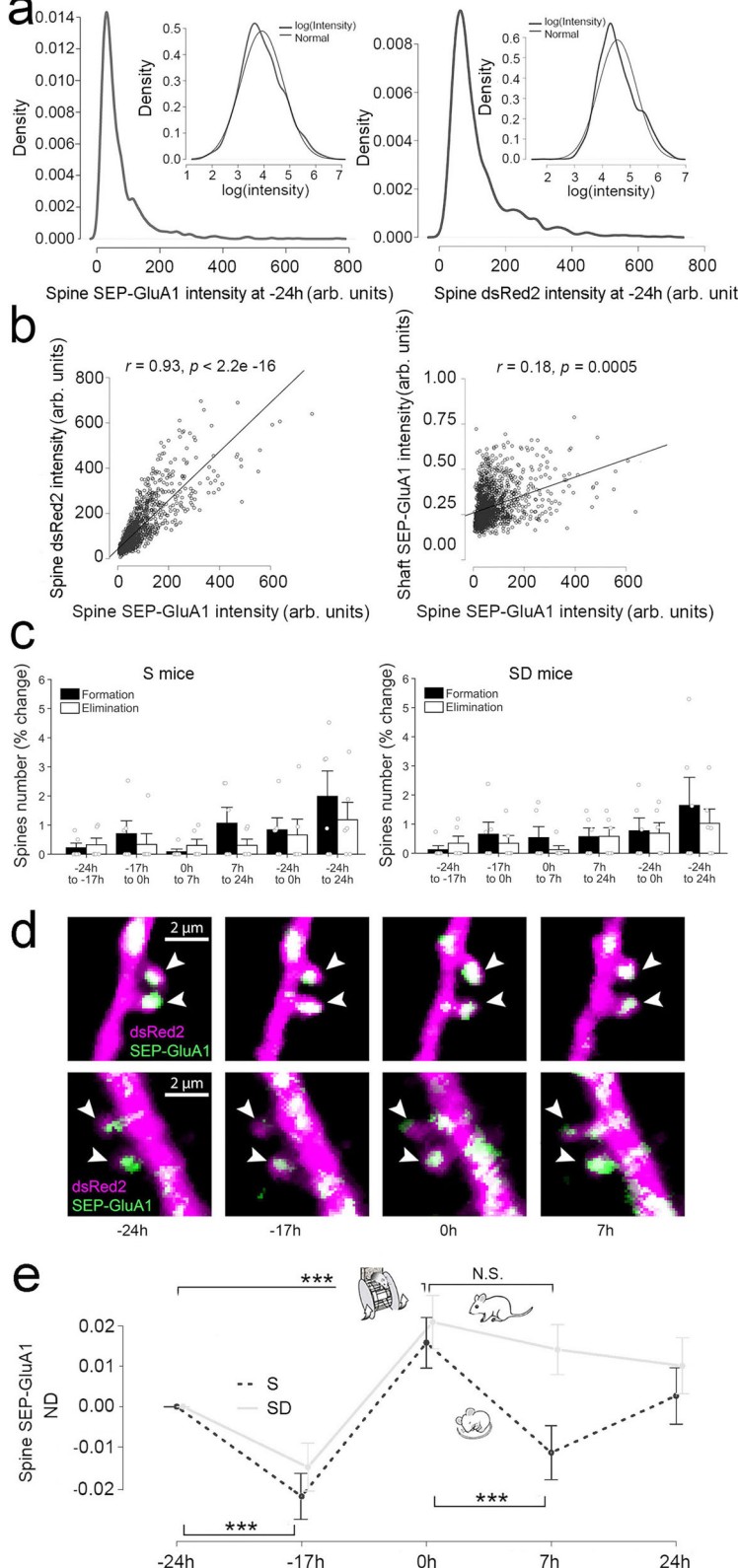

we found a significant decrease in the mean SEP-GluA1 expression after sleep (−24 h > −17 h, $p = 0.00003$, Fig. 2d, e; Supplementary Table 3). We next subdivided all spines in 3 groups—up, down, no change—based on their ND after sleep as compared to before sleep. Using a threshold of ±0.15 ND (equivalent to a post/pre ratio >135% and <74%, respectively), we found that 17.9% of spines (per mouse) were down after sleep, significantly more than

the percentage of spines that were up (12.8%; paired $t$-test; $p = 0.0120$) (Fig. 3a). Similar results were found when other ND thresholds were used (±0.2 ND, 10.9% down and 7.7% up, paired $t$ test; $p = 0.0386$; ±0.1 ND, 26.5% down and 18.8% up, paired $t$-test; $p = 0.0048$; Supplementary Table 4). We also classified each of the 56 dendritic branches as up, down, or same, based on the number of up or down spines that it contained, to determine

**Fig. 2 Mean changes in spine number and spine-surface expression of GluA1 after sleep/waking and motor training. a** Log-normal distributions of the intensities of spine SEP-GluA1 (left) and spine dsRed2 (right) before sleep (−24 h; 12 mice; 1530 spines; 104–192 spines/mouse). Insets, same on a log scale. SEP-GluA1 and dsRed2 intensities were calculated as detailed in Supplementary Fig. 2. **b** Correlations between the intensities of spine SEP-GluA1 and spine dsRed2 ($r = 0.93$, $p < 2.2e{-}16$) or shaft SEP-GluA1 ($r = 0.18$, $p = 0.0005$) before sleep (−24 h). arb. units = arbitrary units. Mean correlation per mouse, significance tested using two-sided one-sample $t$ test. **c** Spines newly formed and eliminated during the indicated time intervals (mean ± SEM). White circles indicate individual mice (6 S mice, 6 SD mice). S, sleep; SD, sleep deprivation. **d** Two representative examples (from 12 mice in total) of repeated in vivo two-photon imaging of layer 2/3 pyramidal cell apical dendrites and their spines (arrowheads). SEP-GluA1 (green), dsRed2 (red), overlap (white). **e** Normalized difference (ND, mean ± SEM) of spine SEP-GluA1 expression relative to −24 h (12 mice; 1530 spines). We show ND instead of the SEP-GluA1 expression because these are the relevant error bars for statistical testing. Two-sided likelihood ratio test followed by two-sided post-hoc comparisons: pre-learning sleep ***$p = 0.00003$ ($n = 12$); motor learning ***$p = 0.00004$ ($n = 12$); post-learning sleep ***$p < 2.2e{-}6$ ($n = 6$); post-learning sleep deprivation, $p = 0.202$ ($n = 6$). Source data are provided as a Source data file.

whether the described changes in spines were restricted to some dendritic branches (4-6 dendrites/mouse; 27.3 ± 14.2 spines/dendrite, mean ± std). Independent of ND threshold, after sleep there were roughly twice as many branches down as there were branches up, and together they accounted for the great majority of dendrites (±0.15 ND: 31 down and 17 up; ±0.2 ND: 30 down and 18 up; ±0.1 ND: 37 down and 13 up) (Fig. 3b). Spines were also subdivided in quintiles and ranked in strength based on the average between post-sleep SEP-GluA1 expression (−17 h) and pre-sleep SEP-GluA1 expression (−24 h). Rankings were calculated on the mouse level and based on the average of −24 h and −17 h (as opposed to solely −24 h) to mitigate any potential effects of regression to the mean (Supplementary Fig. 3)[32]. We found that both up and down spines could occur in any quintile (Fig. 3c).

The log-normal distribution of spine SEP-GluA1 expression present before sleep was maintained after sleep (−24 h and −17 h, Supplementary Fig. 4a). Because log-normal distributions are thought to emerge from multiplicative dynamics[18], we tested whether the changes in SEP-GluA1 expression after sleep were proportional to the expression of SEP-GluA1 before sleep, and found that this was the case (Supplementary Fig. 4b). Consistent with the result reported by[22], spine SEP-GluA1 expression before sleep (−24 h) was also negatively correlated with the ratio between GluA1 levels after sleep and GluA1 levels before sleep (Supplementary Fig. 4b', left panel). The changes in spine SEP-GluA1 intensity after sleep were positively correlated with the changes in the intensity of spine dsRed2, but not with the changes in the intensity of shaft SEP-GluA1 (Supplementary Fig. 5a).

**Learning leads to a net increase in SEP-GluA1 expression.** We then applied an LME model to assess the effects of motor learning, using again time as a categorical fixed effect, and spine, dendrite and mouse as random effects (see "Methods"). SEP-GluA1 expression in the spines was compared across 24 h, the first day without motor learning and the next day immediately after the first training session in the complex wheel task (−24 h to 0 h). Mice were compared at the same time of day (light onset) to rule out effects due to time of day and sleep/waking behavior. We found a significant increase in mean SEP-GluA1 expression of spines after motor learning (ND 0 h > −24 h, $p = 0.00004$, Fig. 2e; Supplementary Table 3). Opposite to the results after pre-learning sleep, the percentage of spines up after training significantly outnumbered that of the spines down (±0.15 ND, 19.5% up and 13.4% down, $p = 0.0097$; ±0.2 ND, 13.5% up and 8.9% down, $p = 0.0006$; ±0.1 ND, 28.4% up and 22.4% down, $p = 0.0476$; Supplementary Table 4) (Fig. 3d). There were also roughly twice as many dendritic branches up as there were down, and together they accounted for most branches (±0.15 ND: 15 down and 33 up) (Fig. 3e), suggesting that the effects of motor learning, like those of sleep, were widespread. Up and down spines occurred in all quintiles (Fig. 3f).

The changes in SEP-GluA1 expression after training were proportional to the expression of SEP-GluA1 before training, in line with the results before and after sleep (Supplementary Fig. 4c). Moreover, the changes in spine SEP-GluA1 intensity after learning were positively correlated with the changes in the intensity of spine dsRed2 (Supplementary Fig. 5b).

**A net, post-learning decrease in SEP-GluA1 occurs only with sleep.** Next, we assessed changes in AMPA receptor plasticity and spine size when training in the complex wheel task was followed by sleep or sleep deprivation. Specifically, we asked whether the sleep-dependent decline in mean SEP-GluA1 expression observed in the spines during pre-training sleep also occurred in mice that slept after the first training session, but not in mice that were kept awake. For this analysis, we compared the third imaging sessions immediately after training (0 h), with the fourth imaging session 7 h later (7 h) (Fig. 2d, right panels). After the 0 h measurement, half of the mice were allowed to sleep (S), while the other half were kept awake (SD). We found a significant interaction between time and condition (likelihood ratio test: $p = 0.0238$; Supplementary Table 3). Post-hoc comparisons showed that mean SEP-GluA1 expression in the spines decreased after sleep but not after sleep deprivation (ND 0 h > 7 h, S mice, $p = 2.2e{-}6$; SD mice, $p = 0.202$; Fig. 2e). After post-learning sleep the number of down spines significantly outnumbered that of up spines (±0.15 ND, 20.3% down and 12.3% up, $p = 0.0349$; ±0.2 ND, 12.4% down and 7.7% up, $p = 0.0810$; ±0.1 ND, 30.2% down and 19.9% up, $p = 0.0450$; Supplementary Table 4) (Fig. 3g). Similar results were found at the dendrite level (Fig. 3h) and both up and down spines could occur in any quintile (Fig. 3i). By contrast, when sleep deprivation followed motor training, a similar number of spines went up or down (±0.15 ND, 15.6% down vs. 13.5% up, $p = 0.3242$; ±0.2 ND, 8.9% down vs. 8.2% up, $p = 0.6754$; ±0.1 ND, 25.6% down vs. 23.5% up, $p = 0.5704$; Supplementary Table 4) (Fig. 3j). Similar results were found at the dendrite level (Fig. 3k). Up and down spines were present in all quintiles (Fig. 3l). In both conditions (S, SD) the changes in spine SEP-GluA1 expression were proportional to the expression of spine SEP-GluA1 immediately after training (Supplementary Fig. 4d, e). Also, the changes in spine SEP-GluA1 expression correlated with the changes in the expression of spine dsRed2 (Supplementary Fig. 5c, d).

Only a minority (24%) of the spines that changed after post-learning sleep had also changed after pre-learning sleep (±0.15 ND; spines up 16/173 = 9.2%; spines down 39/267 = 14.6%). To compare the two types of sleep more directly we fitted an LME model with sleep, training, and their interaction as fixed effects (see "Methods"; before sleep = −24 h, 0 h; after sleep = −17 h, 7 h; no training = −24 h, −17 h; training = 0 h, 7 h) and tested for an interaction to determine if the overall effect of sleep was the same with or without training. We found no significant interaction ($p = 0.7167$; Supplementary Table 3), indicating that

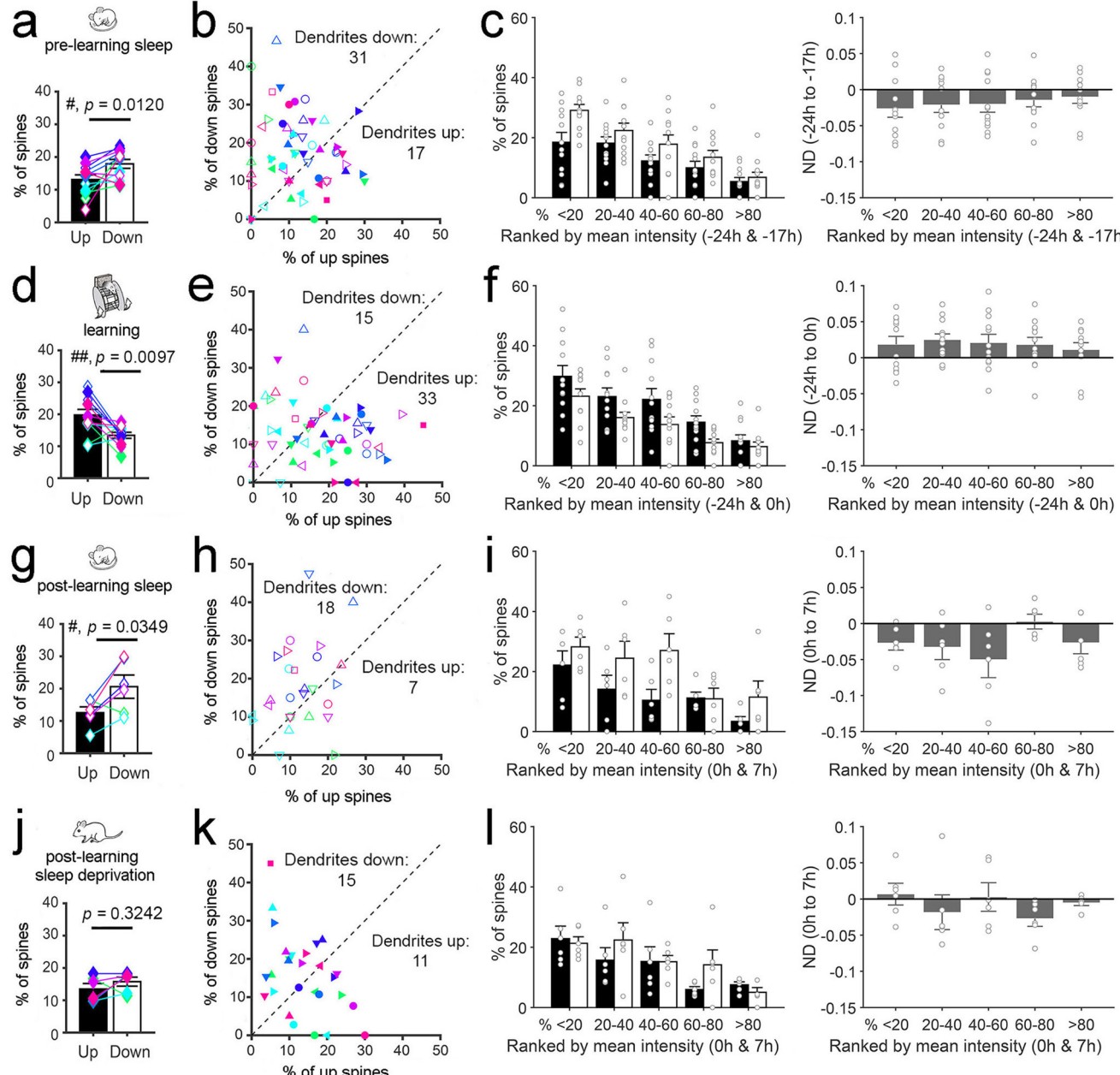

**Fig. 3 Single spine analysis of changes in SEP-GluA1 expression after sleep/wake and motor training. a, b** Percentage of up (ND > 0.15) and down (ND < −0.15) spines (**a**) and up and down dendrites (**b**) after pre-learning sleep. In **a** colored symbols indicate individual animals (n = 12 mice; mean ± SEM). **c** Left, percentage of up spines (black bars) and down spines (white bars) in each quintile; right, ND (normalized difference; mean ± SEM) in spine SEP-GluA1 expression in each quintile. Spines were subdivided in quintiles and ranked in strength based on the mean SEP-GluA1 expression of the two time points indicated on the x axis. White circles indicate individual animals (n = 12 mice). **d–f** Same as (**a–c**) for motor learning (n = 12 mice). **g–i** Same as (**a–c**) for post-learning sleep (n = 6 mice). **j–l** Same as (**a–c**) for post-learning sleep deprivation (n = 6 mice). In **a, d, g, j** the p values are computed using a two-sided paired sample t test. In **b, e, h, k** the percentage of up/same/down spines in each dendritic branch is as follows (mean ± std, in %): pre-learning sleep (−24 h to −17 h, all mice): up 12.3 ± 8.16, same 70.3 ± 12.2, down 17.4 ± 9.9; learning (−24 h to 0 h, all mice): up 17.9 ± 10.9, same 68.7 ± 11.9, down 13.4 ± 7.62; post-learning sleep (0 h to 7 h, 6 mice): up 12.4 ± 6.7, same 68.7 ± 15.4, down 18.9 ± 12.3; post-learning SD (0 h to 7 h, 6 mice): up 13.8 ± 7.31, same 70.5 ± 9.7, down 15.7 ± 10.4. Source data are provided as a Source data file.

despite affecting largely different spines, the overall effect of sleep was the same with or without training.

**The post-learning decrease in SEP-GluA1 is linked to offline consolidation.** We focused next on max spines, defined as those that underwent the largest increase in SEP-GluA1 expression after learning (ND > 0.2 between 0 h and −24 h; p = 6.5e−10; max spines as % of all spines = 13.5 ± 4.1%, 207/1530; mean ± sd;

range per mouse = 6.3–20%; Supplementary Fig. 6), while all the remaining spines were defined as "other" spines (Supplementary Fig. 6). The underlying assumption was that at least some of the max spines are related to learning. In line with this, the max spines maintained high SEP-GluA1 levels 7 h as well as 24 h after the first training session, our last imaging timepoint (Fig. 4a). In the other spines, by contrast, SEP-GluA1 levels did not change in SD mice but decreased in S mice at 7 h as compared to −24 h

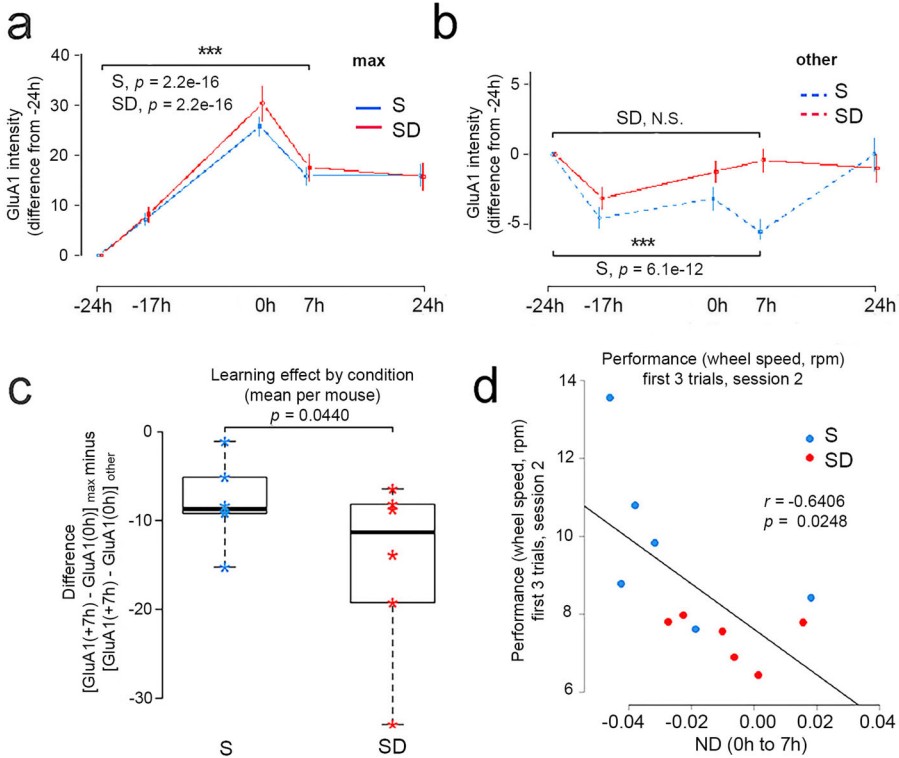

**Fig. 4 Post-training changes in max and other spines. a** SEP-GluA1 intensity in max spines (difference from −24 h) across all time points (mean ± SEM). The max spines are defined as those that showed the largest increase from −24 h to 0 h (max spines as % of all spines = 13.5 ± 4.1%, 207/1530; mean ± sd; range per mouse = 6.3–20%; see also Supplementary Fig. 6). These spines maintain high SEP-GluA1 levels at later times (7 h vs −24 h; p = 2.2e−16 in both groups). S, sleep (n = 6 mice); SD, sleep deprivation (n = 6 mice). **b** SEP-GluA1 intensity in the other spines (difference from −24 h) across all time points (mean ± SEM). Note the different scale of the Y axes in (**a**, **b**). During the 7 h after the first training session, the other spines of S mice show a decrease at 7 h vs −24 h (p = 6.1e−12), while the other spines of SD mice do not (7 h vs −24 h, N.S.). In **a**, **b** statistical analysis was performed using linear mixed effect models, with session as a categorical fixed effect, and spine, dendrite and mouse as random effects. A likelihood ratio test was used to check for an effect of session. S, sleep (6 mice); SD, sleep deprivation (6 mice). **c** The difference between the change in SEP-GluA1 from 0 h to 7 h in the max spines and the change in SEP-GluA1 from 0 h to 7 h in all other spines. There is a significant effect of group (S or SD; likelihood ratio test, p = 0.0440), indicating an advantage for the max spines and a penalization of all other spines in the S group (6 mice) relative to the SD group (6 mice). Center line indicates the median, the box boundaries indicate the first and third quartiles and the whiskers indicate the minimum and maximum values. **d** Negative correlation between performance at the onset of session 2 (first 3 trials) and ND (normalized difference; mean ± SEM) in spine SEP-GluA1 expression in each mouse between 7 h (after S or SD) and 0 h (after training). All spines (r = −0.6406; two-sided Fisher's z transformation, p = 0.0248). Negative ND values (net decrease in SEP-GluA1 expression) are associated with better performance. S, sleep (6 mice); SD, sleep deprivation (6 mice). Rpm, revolutions per minute. Source data are provided as a Source data file.

(Fig. 4b). To further test the hypothesis that max spines remain potentiated after learning while fully correcting for regression to the mean effects, we fitted an LME model with time (7 h and 24 h), learning, and their interaction as fixed effects, and dendrite as a random effect. We found no significant interaction between time and learning (Supplementary Table 3), indicating that the difference between max and the other spines persisted at 7 h and 24 h. Finally, to further test the link between max spines and learning we took advantage of a minimal training control. We used the second training session, in which mice performed the same number of trials but there was limited new learning (Fig. 1d). Consistent with some of the max spines being linked to learning, there were significantly fewer spines with ND > 0.2 between 0 and +24 h (second training session) than between −24 h and 0 (0–24 h = 154, 10% vs −24 h-0 = 207, 13.5%; p = 0.0386).

The max spines remained potentiated after the first training session despite losing some SEP-GluA1 in the first 7 h after learning (Fig. 4a). These changes occurred in both S mice and SD mice, indicating that sleep has no specific effect on the max spines. In the other spines, however, SEP-GluA1 levels decreased

after the first training session if mice were allowed to sleep but not if they were kept awake (Fig. 4b). Thus, sleep may still provide a relative advantage to the max spines by weakening all the remaining spines. To test this hypothesis we computed the difference in SEP-GluA1 expression between time 0 h (immediately after learning) and time 7 h (after S or SD) for each spine and fitted an LME model for this difference with condition, learning, and their interaction as categorical fixed effects, and dendrite and mouse as random effects. Using a likelihood ratio test, we found a significant interaction between condition and learning (p = 0.04396; Supplementary Table 3). Specifically, after sleep max spines were, in relative terms, in a stronger position than after sleep deprivation, because the expression of SEP-GluA1 decreased in the other spines after sleep but not after sleep deprivation (Fig. 4c). Note that the decrease in SEP-GluA1 expression in the max spines (Fig. 4a) may have been due, at least in part, to regression to the mean effects, even if we applied a correction for these effects[33]. Crucially, however, the significant interaction that we found was not confounded by any residual effect due to regression to the mean, which would affect the sleep and the sleep deprivation condition to the same extent.

We then asked whether the relative advantage afforded by sleep applied to all max spines, or to those more likely to be related to learning. We found that the relative advantage was also present in the subset of spines with ND > 0.2 both between 0 h and −24 h and between 0 h and −17 h (likelihood ratio test, $p = 0.0328$; Supplementary Table 3). This was not true for the subset of spines with ND > 0.2 between 0 h and −17 h but not between 0 h and −24 h (likelihood ratio test, $p = 0.84$; Supplementary Table 3). In this subgroup the increase in SEP-GluA1 expression may be linked to circadian, stochastic, or other factors rather than to learning.

Finally, we found that performance at the beginning of the second session, a measure of offline consolidation, was negatively correlated with the change in SEP-GluA1 expression between 0 h and 7 h, when considering all spines ($r = −0.6406$; $p = 0.0248$; Fig. 4d) or just the other spines ($r = −0.5946$, $p = 0.0414$; max spines: $r = −0.3924$, $p = 0.2071$). Thus, performance was low in mice that showed a net increase in spine SEP-GluA1 expression at 7 h (positive ND at 7 h) and high in mice with a net decrease in spine SEP-GluA1 expression (negative ND at 7 h).

**Up spines during sleep are best explained by spontaneous periods of wake**. Our results show that, for a majority of spines, SEP-GluA1 expression went up during wake and down during sleep. However, a minority of spines showed an increase of SEP-GluA1 expression during nominal sleep periods. We note, however, that the percentage of wake during these nominal sleep periods was on average 36%, higher than in our previous study[21], and possibly a consequence of the experimental paradigm that required the mice to undergo several repeated imaging sessions across 48 h. Therefore, it is an open question whether the up spines observed after the nominal pre-learning sleep period should be attributed to the majority of sleep epochs or to the intervening minority of wake epochs, especially if these occurred during the last few hours before imaging. To try to distinguish between these possibilities we evaluated two different models for the effects of sleep. Fitting the models took into account the proportion of spines that increased (ND > 0.15) and decreased (ND < −0.15), ensuring that the models capture the behavior of individual spines and not just population level statistics. The first model was a sleep-dependent potentiation model, according to which some spines are downscaled during sleep while others are upscaled, combined with an additive independent noise source. This model captures the hypothesis that an increase in SEP-GluA1 intensity of some spines is a specific effect of sleep. The second model was an intervening-wake potentiation model, according to which spines are downscaled during sleep, but some spines are upscaled due to intervening wake epochs, again combined with an additive independent noise source. Using a similar number of free parameters (5 for the sleep-dependent potentiation model; 4 for the intervening-wake potentiation model), the two models accounted equally well for the mean, variance and correlation of the SEP-GluA1 intensities before and after sleep (−24 h to −17 h; 3.39% relative error for the sleep-dependent potentiation model, 3.26% relative error for the intervening-wake model; Supplementary Table 5). However, the intervening-wake model accounted better for upscaling and downscaling behavior in each quintile (relative error, 2.22% for wake-noise model; 4.32% for upscaling model) (Supplementary Fig. 7). Specifically, while both models replicated the overall proportion of spines that scaled up or down, the sleep-dependent potentiation model overestimated the number of small spines that scaled and underestimated the number of large spines that scaled (Supplementary Fig. 7).

## Discussion

In this study, we assessed the effects of learning and sleep on the strength of excitatory glutamatergic synapses by measuring the surface expression of GluA1-containing AMPA receptors on dendritic spines in the superficial layers of the mouse primary motor cortex. Motor training resulted in a net increase in spine-surface GluA1 expression, consistent with a net increase in synaptic strength, while sleep had the opposite effect. Both the average change in spine SEP-GluA1 expression and the percentage of spines undergoing learning-related and sleep-related changes are comparable to what was reported in previous studies that measured the same marker in response to sensory experience, motor training, and behavioral state[22,34,35]. Moreover, the upregulation of spine SEP-GluA1 levels after training in the complex wheel task is in line with previous work showing that the acquisition of motor skills leads to long-term potentiation of cortical connections, to the formation and enlargement of spines, as well as to increased firing of task-related neurons in primary motor cortex[19,25,35–38]. After training, as after sleep, the changes occurred in most dendritic branches, but we do not know whether these results would apply to the whole dendritic tree, nor do we know whether they would extend to the deep layers.

The net decrease in spine-surface GluA1 expression after sleep, on the other hand, is consistent with a net decrease in synaptic strength and in line with ultrastructural evidence. Specifically, in a previous study using serial block-face electron microscopy, we reconstructed ~7000 synapses in primary motor and sensory cortex of adolescent mice (postnatal day 30) and showed that the axon–spine interface, the direct area of contact between presynapse and post-synapse, decreases on average by 18% after 6–8 h of sleep relative to 6–8 h of either spontaneous waking or sleep deprivation using novel objects[21]. A subsequent study in which we reconstructed 3750 spines in primary motor cortex of mouse pups (postnatal day 13) found a net average decline of more than 30% in the size of the axon–spine interface after sleep relative to sleep deprivation[39]. Since structural and functional plasticity are correlated[15,20], these ultrastructural findings strongly suggested that in the immature and adolescent cortex most synapses weaken after several hours of sleep. The present results confirm the ultrastructural demonstration that sleep leads to a net reduction of synaptic strength using selective molecular markers and extend it to the adult primary motor cortex.

We also found no net change in spine SEP-GluA1 levels between time 0, immediately after motor training, and the end of the sleep deprivation 7 h later, suggesting that the exposure to novel objects could not induce further synaptic strengthening in the area where learning-induced synaptic potentiation had just occurred. Previously, we had found that in the rat barrel cortex long-term potentiation could be induced by electrical stimulation after a few hours of sleep but not after a few hours of sleep deprivation[40]. After short periods of sleep deprivation the induction of long-term potentiation is often impaired, while the induction of long-term depression is spared or promoted (reviewed in ref. [41]). Together, these results point to a limited daily capacity of the cortex to undergo synaptic strengthening unless sleep occurs.

We have proposed that the reestablishment of synaptic homeostasis through synaptic down-selection may be a central function of sleep, which may be thus considered as 'the price' the brain pays for plasticity during wake[6]. So far, sleep-dependent synaptic renormalization had been demonstrated following periods of wake in enriched environments that promote plasticity throughout the brain[21,39,40]. However, sleep-dependent synaptic renormalization should also occur in the cortical circuits involved in learning a specific task. Here we tested this prediction, by focusing on the superficial layers of primary motor cortex, where

plastic changes in response to motor skill learning are well characterized[25,31,35–38]. We found that the spine-surface expression of the GluA1-containing AMPA receptors does in fact increase in the superficial layers of primary motor cortex after training in the complex wheel task, as was also the case after learning a reaching task[35]. Crucially, we also found that the spine-surface expression of the GluA1-containing AMPA receptors declines during post-learning sleep and does so to an extent comparable to that occurring when sleep is not preceded by motor training. Thus, at least in the superficial layers of primary motor cortex, the results are consistent with sleep leading to an overall decline in synaptic strength also in networks that are directly engaged in learning. Whether this result generalizes to other layers or areas of the adult brain is unknown, but we predict that the need to rebalance overall synaptic strength during sleep should apply to all synapses undergoing a net increase in synaptic strength due to learning. Note also that we assessed the overall, net effect of several hours of sleep but could not determine whether NREM sleep and REM sleep played similar, different, or even opposite effects on synaptic strength, as suggested[8].

In these experiments, both the number and the strength of the spines could be measured with single spine resolution. At each time point there was a balance between spine formation and elimination, consistent with the previous evidence[42]. Notably, spine density did not change 24 h after learning, or after 7 h spent mainly asleep or awake. On the other hand the strength, as indexed by the spine-surface expression of GluA1-containing AMPA receptors, increased after learning and decreased after sleep. Furthermore, these changes were positively correlated with changes in spine size, as measured by the structural marker dsRed2 (in line with previous studies using the same markers[34,35]. Thus, sleep-dependent plastic changes in the excitatory synapses of the adult cortex were mediated by changes in spine size and SEP-GluA1 expression, but not in spine number. By contrast, in the adolescent cortex spine formation and spine elimination happen at all times, independent of behavioral state, but sleep and wake significantly bias spine turnover, with more spines being eliminated than formed after several hours of sleep, and the other way around after wake[43,44].

Another recent study applied two-photon laser-scanning microscopy to track spine SEP-GluA1 expression in the primary cortex of young adult mice[22]. In that study, GluA1 expression was measured for 3 consecutive days after the first 4 h of the dark phase, when mice are likely to be awake, and after the first 4 h of the light phase, when mice are likely to be asleep, yielding an average light/dark ratio in SEP-GluA1 expression. The ratio was smaller than one in 58% of the 383 spines that were analyzed and greater than one in the remaining 42%[22], in line with the current results that during sleep the spines that lost some surface GluA1-containing AMPA receptors outnumbered those that showed the opposite change. However, this study did not attempt to control for the contribution of regression to the mean and for the role of wake periods in accounting for the up spines. In the same study, a negative correlation was observed between spine-surface GluA1 expression during wake and the sleep/wake ratio in GluA1 levels. A negative correlation was also present in our data, but with no indication of size-dependence and even for completely independent observations (Supplementary Fig. 3b', right panel). In other words, our data do not support the hypothesis that the sleep-dependent decline in SEP-GluA1 expression occurs mainly or exclusively in the largest spines. To the contrary, small spines were more likely to change than big spines, both after learning and after sleep. This result is consistent with ultrastructural findings in adolescent mice, in which the sleep-dependent decrease in the size of the axon–spine interface occurred in small and medium size synapses but not in the largest ones[21].

Numerous previous studies have also reported that small spines change their shape and size quickly, while large mushroom spines are more stable[15,45,46].

Sleep-dependent synaptic down-selection may offer a relative advantage to the subsets of synapses involved in learning due to synaptic tags or a more coherent reactivation during sleep, which may allow these synapses to escape the overall downregulation[6,14]. Another possibility is that the reactivation of synapses causally involved in learning during wake leads to their further strengthening during sleep[10–12]. A direct test of these mechanisms is challenging, because it requires the ability to identify task-related synapses and measure changes in their strength in vivo after learning and after sleep. Here we attempted to address this point by focusing on the max spines, defined as those whose levels of surface GluA1 expression increased the most immediately after the first training session. We assume that at least a subset of them was specifically linked to skill learning. This assumption is supported by the findings that the max spines remained potentiated after learning, as indicated by high levels of SEP-GluA1 24 h after the first training session. Furthermore, the difference in SEP-GluA1 expression between max and other spines, which was present immediately after learning, persisted 24 h later, after the second training session. Also, compared to the first training session, there were fewer max spines after the second training session, when little additional learning occurred. Because this was our last imaging timepoint, we do not know for how long max spines maintained high GluA1 expression. In a recent study using the reaching task, SEP-GluA1 levels increased in 8.3% of all spines and remained elevated 1 week after the last day of training[35]. Of note, we also found that the relative advantage afforded by sleep only applied to the max spines more likely to be involved in learning. However, more direct evidence is needed to prove that max spines are specifically linked to learning, and that the effects of sleep are specific to the learned spines. Note that the levels of SEP-GluA1 in the max spines decreased to some extent after both sleep and sleep deprivation but, even after applying a statistical correction[33], it is possible that regression to the mean effects contributed to this decrease (spines that went up the most are more likely to go down). Crucially, however, after sleep the max spines were, in relative terms, in a stronger position than after sleep deprivation, because the other spines lost some GluA1 only after sleep. This result is not confounded by any residual effect due to regression to the mean, which would affect the sleep and the sleep deprivation condition to the same extent. Finally, the analysis of max and the other spines assumes the presence, in both spine groups, of a linear relationship between the intensity of the SEP signal and the expression levels of SEP-GluA1. While this is difficult to prove, there is no reason to think that any deviation from linearity, if present, would differentially impact the S and SD groups.

At the beginning of the second training session all mice remembered well how to perform the task. However, mice allowed to sleep performed significantly better than sleep deprived mice, consistent with previous findings in rodents and humans[5]. Max spines are likely to reflect successful skill learning, and indeed they remained strong post-learning in both S and SD mice. On the other hand, max spines did not differ between the two groups, and their level was not correlated with performance at the beginning of the second training session, suggesting that they are not directly involved in the performance advantage afforded by post-learning sleep. Conversely, good performance in the second session correlated with the downregulation of SEP-GluA1 after the first learning session in all spines or in the other spines, which were the great majority of spines. Together, these findings suggest that sleep benefits memory consolidation by increasing the signal-to-noise ratio, and does so by decreasing the

noise (other spines) rather than by increasing the signal per se (max spines). In other words, the advantage provided by sleep seems to depend on its weakening effect on the other spines, rather than on a direct effect on the max spines. In a rat model of neuroprosthetic learning, the firing of a small group of task-related neurons whose activity was causally linked to learning the task increased slightly after post-learning sleep, whereas firing decreased substantially in a larger set of task-unrelated neurons[47,48]. Future experiments in which both neuronal and synaptic activity are measured in the same animal after learning and post-learning sleep could in principle determine whether the weakening effect of sleep on the other spines can account for the effects on neuronal firing. Similar experiments could also clarify why visual deprivation leads to an early, acute drop in the firing of cortical neurons followed by a slow recovery phase, during which a net increase in neuronal activity is confined to the wake period and does not occur during sleep[49].

Previous studies have shown that although newly formed spines are in general significantly less stable than pre-existing spines, they are more likely to stabilize after motor learning, especially when training continues for several days[37,50]. In young mice the percentage of new spines formed after a single training session correlated with the number of successful reaches during the same session[37]. Moreover, in both young and adult mice the percentage of new spines that persisted after 7–10 days of repetitive training correlated with skill retention and performance improvement over the course of training[50,51]. Together, these studies were able to link the formation of new spines with learning and skill refinement, but did not test for any specific role of sleep. Other experiments found that the loss of NREM sleep after motor training reduced spine formation[17], and the post-training loss of REM sleep decreased the elimination of new spines[52], but did not test whether the sleep-dependent changes in spine turnover correlated with the changes in task consolidation. Here we confirmed that sleep promotes the consolidation of the complex wheel task and found that this effect depends on the extent to which the other spines are weakened during post-learning sleep. We speculate that, because of their relative advantage, the max spines may have been in the best position to undergo further strengthening, as indexed by SEP-GluA1, during the second training session.

While the majority of spines showed a decrease in SEP-GluA1 expression during pre- and post-learning sleep periods, a minority of spines showed an increase in SEP-GluA1 expression. These up spines may result from the sleep-dependent potentiation of a select group of synapses. Alternatively, since mice were awake on average 36% of the time during the 7 h of sleep opportunity, up spines may result from synaptic potentiation occurring during the intervening periods of wake. Note that the expression of SEP-GluA1 reflects the turnover of surface GluA1-containing AMPA receptors, which can occur rapidly[28,29], hence it may be more sensitive to short periods of wake than, say, ultrastructural synaptic markers such as the axon–spine interface. The current experimental paradigm cannot provide direct evidence in support of either sleep-dependent potentiation or intervening-wake potentiation, because the sleep group was not always asleep. Our statistical modeling showed that the intervening-wake potentiation model matched closely the proportion of both up and down spines and did so across all quintiles. By contrast, the sleep-dependent potentiation model overestimated both up and downscaling in small spines and underestimated scaling in large spines. We recognize, however, that these results cannot be taken as direct evidence in favor of the intervening-wake potentiation model.

The current results support the hypothesis that sleep is needed to reduce overall synaptic strength following learning-induced net synaptic potentiation, raising the question of which homeostatic mechanisms are involved[53]. Several such mechanisms have been identified in response to sensory deprivation, and their contribution may differ depending on the experimental model. For instance, monocular and whisker deprivation trigger early changes in excitation/inhibition followed by global synaptic upscaling[54], while the recovery of synaptic strength after dark rearing occurs through metaplasticity, by sliding the threshold to induce Hebbian, synapse-specific plasticity in favor of long-term potentiation[55]. Global downscaling, in which all synapses decrease proportionally to their size, is an attractive candidate for sleep-dependent synaptic weakening and could result from the neuromodulatory changes that occur during sleep, including low levels of catecholamines, which span the entire brain and promote synaptic depression[6]. However, we found that at the population level, the decrease in synaptic size during sleep is widespread but not global in cortex, sparing on average the largest synapses[21]. More crucially, sleep promotes memory consolidation as well as memory integration and gist extraction, suggesting at least some specificity in its synaptic effects. Such specificity could arise by a process of smart down-selection by which some of the molecular changes that occur during learning and the induction of synaptic potentiation, could make some synapses more resistant to depression. Candidate mechanisms include the phosphorylation of GluA1 at Ser845, which makes these receptors resistant to downscaling[56], and/or of GSK-3beta[57]. Other mechanisms could also help confining synaptic strengthening during wake. For instance, synaptic inhibition is upregulated during the light/sleep period in visual and prefrontal cortex and hippocampal CA1, while synaptic excitation is greater at night, during the major wake period[58]. Moreover, short sleep deprivation also increases the expression of the NR2A subunit of the NMDA receptor, a change that facilitates long-term depression[40,59] and depotentiation[60]. Independent of the specific mechanisms involved, these results demonstrate that sleep-dependent synaptic renormalization is not confined to specific developmental periods but extends to the adult brain.

## Methods

**Animals**. C57BL/6J mice (Jackson Laboratory) were maintained on a 12 h light/12 h dark cycle with food and water available ad libitum (21–23 °C, 30–40% relative humidity). All animal procedures and experimental protocols followed the National Institutes of Health Guide for the Care and Use of Laboratory Animals and were approved by the licensing committee. Animal facilities were reviewed and approved by the institutional animal care and use committee (IACUC) of the University of Wisconsin-Madison and were inspected and accredited by the association for assessment and accreditation of laboratory animal care (AAALAC).

**In utero electroporation (IUE)**. Embryos from timed pregnant C57BL/6 mice were transfected by IUE at embryonic day 14.5 (E14.5), in order to restrict expression to cortical layers 2/3 excitatory neurons. Under isoflurane anesthesia (2% induction, 1–1.5 % maintenance in 100% $O_2$) ~1 µl of plasmid DNA solution containing the marker Fast Green (0.1 µg/µl) was pressure injected into the lateral ventricle in the right hemisphere of each embryo through a pulled-glass pipette. The DNA solution contained SEP (super-ecliptic pHluorin)-GluA1, myc-GluA2, and dsRed2 at a 4:4:1 ratio (2, 2, 0.5 µg/µl). GluA2 was included to approximate the natural GluA1/GluA2 ratio in transfected cells. After the DNA injection, 6 pulses of 25–26 V (50 ms on, 950 ms off at 1 Hz) were delivered, targeting the fronto-parietal cortex, using 5-mm tweezer electrodes connected to a square wave electroporator (CUY21, π Protech).

**Immunostaining**. Adult mice (3–4 months old) with expression in primary motor cortex were transcardially perfused under deep anesthesia (2% isoflurane in oxygen) with 4% paraformaldehyde (PFA) with a 24-h post-fix in PFA. Brains were subsequently sectioned with a vibratome (Leica) into 40 µm thick coronal sections. Sections were washed with PBS (pH 7.4) and treated with a blocking solution (PBS with 5% NGS, 1% BSA) for 1 h, incubated overnight in the blocking solution containing mouse anti-GluA1 antibody conjugated to Alexa Fluor 647 (1:50, Santa Cruz Biotechnology, sc-55509 AF647) at 4 °C, washed three times with PBS, mounted, and air-dried. 512 × 512 pixels (298 µm × 298 µm) Z-stack images with 1 µm Z-step were acquired with a confocal microscope (Olympus BX61W1, objective lens; UPlanFL N40x, NA 1.30). Based on Alexa 647-GluA1 signal (700/75

emission filter), regions of interest (ROIs) were manually drawn for 28 SEP-GluA1 positive (525/70 emission filter) cell bodies and 28 nearby SEP-GluA1 negative cell bodies. Average intensity was summed across best ±1 Z planes (Supplementary Fig. 1).

**Sleep/waking behavior and sleep deprivation.** Sleep and waking states were determined by continuous monitoring with infrared cameras (OptiView Technologies) starting at least 24 h before the first training session. This method cannot distinguish NREM sleep from REM sleep, but it consistently estimates total sleep time with ≥90% accuracy[43]. Motor activity was quantified by custom-made video-based motion detection algorithms (Matlab)[61]. Sleep deprivation was enforced by exposure to novel objects, in which toys and other objects of different shape, color, and texture were introduced in the cage. Mice were stimulated only when they appeared drowsy, assumed a typical sleeping position, and/or closed their eyes. Mice were never disturbed when they were spontaneously awake, feeding or drinking. Mice were adapted to the procedure for several days before the experiment. It is well established that this method almost completely abolishes sleep for up to 7 h, with mice spending asleep only 1–5% of the entire sleep deprivation period[31,62,63].

**Complex wheel task.** The complex wheel task was previously optimized in our laboratory[31]. Specifically, we used a modified accelerating rotarod system (EZRod, Omnitech Electronics, Inc.) equipped with a wheel in which 20 of the original 50 rungs have been removed, to generate 2 identical complex sequences of rungs in one rotation. Mice were handled for 3 days in the room used for two-photon microscopy (1–2 min/day) before receiving two sessions of training spaced 24 h apart. Mice did not receive any habituation or pretraining using the complex wheel or a regular wheel. Each session included 20 trials (with a 5 min rest after the first 10 trials) and occurred during the last hour of the dark period. At the beginning of each session the mouse was placed onto the stationary complex wheel, and acceleration increased until the mouse fell off the wheel (0–40 rpm in 10 min; acceleration = 223.3 cm/min$^2$). Time and speed when mice fell off the wheel were recorded and used to assess performance. After the first 10 trials, mice were returned to their home cage for a 5-min rest period, during which they mainly groomed but never slept. At the end of the first session mice were imaged using two-photon microscopy and then returned to the home cage and either allowed to sleep, or sleep deprived for 6 h. The second training session occurred 24 h later.

**Craniotomy and two-photon laser scanning microscopy.** Pups born after in utero electroporation (males and females) underwent stereotaxic surgery at the age of P56–P78 under isoflurane anesthesia (2% induction, 1–1.5% maintenance in 100% O$_2$) to make a 3 mm diameter craniotomy and implant a round glass coverslip over the primary motor cortex in the right hemisphere. Cranial windows were centered at AP: 0–0.5 mm and LM: 1.8–2 mm from bregma. Acquired imaging windows were centered at AP: 0.62 ± 0.30 mm, LM: 1.47 ± 0.037 mm for S mice and AP: 0.60 ± 0.23 mm LM: 1.47 ± 0.13 mm for SD mice (mean ± SEM, n = 6 mice for each group). A custom-made metal head bar was glued to the skull with dental cement for head-fixation during imaging. Mice were housed individually after surgery. Two to three weeks after surgery, in vivo imaging was performed under isoflurane anesthesia (2% induction, 1.5% maintenance in 100% O$_2$) for ~30 min per session with a two-photon microscope (Ultima, Prairie Technologies/Bruker, Middleton, WI) and a Ti:Sapphire laser (Cameleon Coherent, Coherent Inc., Santa Clara, CA) tuned to the excitation wavelength of 910 nm. Mice were run in separate experiments, alternating between the two experimental conditions (sleep, sleep deprivation) as much as possible. Images were taken using a water immersion ×60 objective (0.8 numerical aperture, Olympus, LUMPlanFI/IR, Tokyo, Japan) at <50 mW laser power at back aperture. Image stacks from the pial surface to a depth of 70–100 μm were acquired at 1024 × 1024 pixels with a voxel size of 0.12 μm in x and y and a z-step of 1 μm. Each plane was scanned 4 times for frame average with 1.2 μs dwell time per pixel. The same target area was imaged in each session. Although we cannot completely rule out that a few of the spiny dendrites that we imaged did not belong to layer 2/3 excitatory neurons, we think this is very unlikely because electroporation was precisely timed in order to target this cell group. Moreover, among the non pyramidal inhibitory neurons, Martinotti cells have the highest number of dendritic spines but their somas are rare in layer 1 and in the most superficial part of layer 2, and their spine density is ~0.3/μm, lower than in pyramidal neurons[64]. The average spine density of the dendritic branches that we imaged was ~0.5/μm.

**Image analysis.** For pre-processing, each z-stack was aligned and registered using the MultiStackReg plugin of ImageJ[65]. As previously described[34], manually marked spine heads using the structural dsRed2 channel were automatically connected to each traced dendrite, using a custom-written Map Manager software (http://robertcudmore.org/) on IGOR Pro (Wavemetrics, Lake Oswego, OR), kindly shared by Dr. Robert Cudmore (University of California, Davis). Spines included in the analysis had to have more than 3 pixels (>0.36 μm) width and protrude from the dendritic backbone by more than 6 pixels (>0.72 μm) and be primarily parallel to the imaging plane. Spines that were present from one time point to the next were categorized as persistent, spines that were present at a given time point but absent

at the preceding time point were categorized as formed and spines that were present at a given time point but not in the next were categorized as eliminated. For the intensity analysis, only spines that could be identified in all sessions were considered. Each spine was assigned to three regions of interests (ROIs): spineROI, shaftROI, and backgroundROI. The spineROI enclosed the spine head and did not include pixels within the radius of the dendritic backbone. The width of each spineROI (0.5–1.0 μm) was manually adjusted to avoid contamination by nearby spines. We carefully examined the z-planes above and below the spineROIs to confirm that the intensity values were not contaminated by signal from spines protruding out of the imaging plane. The shaftROI was constructed from the backbone line and radius of the dendrite, centered on each spine and expanded (±2 μm) along the dendrite. At nearby background region, same-shape ROIs for the spine and the shaft were defined for background subtraction. All ROIs extended in z-direction up and down (±1 plane) from the best imaging plane manually selected based on the dsRed2 channel. To measure the spine intensity for the green/SEP-GluA1 channel, the same three ROIs for each spine were applied to the SEP-GluA1 channel. The backgroundROI for each channel was separately translated in x/y to a nearby region of the image to minimize the background intensity. To compare intensity values across imaging sessions, after background subtraction, the spine SEP-GluA1, spine dsRed2, or adjacent shaft SEP-GluA1 signals were normalized to the mean of the adjacent shaft dsRed2 signal. Representative images shown in the figures were median filtered, upscaled, and contrast enhanced. For 3D visualization, image stacks were acquired at 1024 × 1024 pixels with a voxel size of 0.19 μm in x and y and a z-step of 3 μm. In Fig. 1a, a representative stack was 3D-reconstructed using the 3D viewer plugin[66] in the Fiji image analysis software[67].

**Statistical analysis.** Statistical analysis of SEP-GluA1 intensities was performed using linear mixed effect (LME) models[68] (Supplementary Table 3). For the current work, LME models are preferred over traditional repeated measures ANOVA because of their ability to handle unbalanced designs (e.g., differing numbers of spines sampled from each mouse), and nested random effects (spine is nested in dendrite, which is nested in mouse). However, analysis using repeated measures ANOVA yields similar results throughout. Maximum likelihood estimation of model parameters was performed in R using the lme4 package[69]. Significance of main effects and interaction terms was done using likelihood ratio tests (LRT), and Cohen's $f^2$ was used as a measure of effect size. For significant effects, post-hoc tests with p-values corrected for multiple comparisons were performed using the glht() function in the multcomp package[70]. Analysis was performed using four separate models, and all models used a square-root transformation of the SEP-GluA1 intensity to stabilize variance. After the square-root transformation, diagnostic plots of the residuals were used to validate the normality and constant variance assumptions.

For the first model (effects of sleep or learning), the response variable was the SEP-GluA1 intensity (Supplementary Table 3). The model contained time (with levels −24 h, −17 h, and 0 h) as a categorical fixed effect, and mouse, dendrite and spine as random effects. For these times, there is no difference between the S and SD mice, thus no condition factor was included in the model. In this model we found a significant effect of time (LRT, $p < 2.2e−16$), and post-hoc comparisons found a decrease after sleep (−24 h > −17 h, $p = 0.00003$) and an increase after learning relative to pre-sleep (−24 h < 0 h, $p = 0.00004$) and post-sleep (−17 h < 0 h, $p < 1e−16$).

For the second model (effects of post-learning sleep or sleep deprivation), the response variable was again the SEP-GluA1 intensity (Supplementary Table 3). The model contained time (with levels 0 h and 7 h) and condition (with levels S and SD) as categorical fixed effects, and mouse, dendrite and spine as random effects. We found a significant interaction between time and condition (LRT; $p = 0.0238$), and post-hoc comparisons found a significant decrease for the S condition (0 h > 7 h, $p = 2.2e−6$) but not for the SD condition (0 h = 7 h, $p = 0.202$).

For the third model (max spines), we used the difference in intensities (7 h–0 h) as the response variable. This model had condition (S and SD) and learning (max, ND > 0.2; other, ND < 0.2) and their interaction as categorical fixed effects, and dendrite and mouse as random effects. We found a significant interaction between condition and learning (LRT; $p = 0.0440$). The significance of the interaction term was robust to the choice of threshold, being significant ($p < 0.05$) or at least trending ($p < 0.1$) for most ND thresholds between 0.17 and 0.28 but not below 0.17, possibly because below that ND threshold the spines were not strongly affected by learning. For ND thresholds larger than 0.28 there were too few spines to reliably test the interaction. We found that the interaction was also significant if we defined the max spines to have ND > 0.2 relative to both −17 h and −24 h ($p = 0.0328$), but not if we defined the max spines to have ND > 0.2 relative to −17 h but not −24 h ($p = 0.8411$). To study the prolonged effects of learning, we extended this analysis by including time (7 h–0 h or 24 h–0 h) and its interaction with learning as categorical fixed effects. We found no significant interaction between time and learning (LRT; $p = 0.4164$), suggesting that the effect of learning is maintained after 24 h.

For the fourth model (comparison between pre-learning sleep and post-learning sleep), we used sleep (before sleep, −24 h and 0 h; after sleep, −17 h, 7 h), training (before training, −24 h and −17 h; after training 0 h and 7 h), and their interaction as categorical fixed effects, and dendrite and mouse as random effects. For this model, we only used the spines from the mice in the S condition. We found no

significant interaction between sleep and training (LRT; $p = 0.7167$), suggesting that mean effect of sleep was the same with and without training.

Further statistical analyses (performance data, Supplementary Table 1; correlations, Supplementary Table 2; proportion of spines that change Supplementary Table 4) were performed using paired-sample $t$-tests, independent-sample $t$-tests, and one-sample $t$-tests as appropriate. All tests were two-sided, and normality was assessed graphically using qq-plots with no major departures. For all two-sample tests, Cohen's $d$ was used as a measure of effect size.

**Sleep-dependent potentiation model and intervening-wake potentiation model**. We evaluated two different models for the effects of sleep (−24 h to −17 h) on spine SEP-GluA1 intensity. The first model (sleep-dependent potentiation model) was a selective up/downscaling model with additive independent noise (normally distributed). The model had five parameters: the proportion of spines to upscale, the magnitude of upscaling, the proportion of spines that downscale, the magnitude of downscaling, and the variance of the independent noise. The second model (intervening-wake potentiation model) was a selective downscaling model with two noise sources, additive independent noise and additive size-dependent wake noise. The wake noise was generated by bootstrap sampling from the effect of pure wake (SD mice from 0 h to 7 h); to be size-dependent, the spines are divided into quintiles based on GluA1 intensity and the wake noise is sampled from the corresponding quintile of the SD data. This model has four parameters, the proportion of spines that downscale, the magnitude of downscaling, the variance of the independent noise and the variance of the size-dependent noise.

We used the observed GluA1 intensities at −24 h and −17 h to estimate the parameters of each model. Both models start by simulating a sample of spine SEP-GluA1 intensities at −24 h from a log-normal distribution with the mean and variance set to match the observed data. A sleep mechanism (either selective down/upscaling or selective downscaling with wake noise) is applied to each individual spine at −24 h to create a sample of intensities at −17 h. For each spine, we then compute the ND at −17 h relative to −24 h, and classify the spines as downscaled, upscaled, or no change. To evaluate the fit of the model, in the simulated data we compute the mean and variance of the intensities at −17 h, the correlation between spine intensity at −24 h and −17 h, and the proportion of spines that up/downscale (based on a threshold of ND = 0.15). For each quantity, we compare the value obtained by simulation with the value in the real sample by computing the relative error (relative error = (true value − estimated value)/true value) and then sum the relative errors. The parameters of each model were tuned numerically to minimize the total relative error of the model (Supplementary Table 5). Note that, by considering the proportion of spines that up/downscale, we ensure that the model captures the behavior of individual spines, not just simply population mean and variance.

**Reporting summary**. Further information on research design is available in the Nature Research Reporting Summary linked to this article.

## Data availability
Source data are provided with this paper. All relevant data are available from the authors upon reasonable request.

## Code availability
Image analysis was performed using a custom-written Map Manager software (http://robertcudmore.org/) on IGOR Pro (Wavemetrics, Lake Oswego, OR), kindly shared by Dr. Robert Cudmore.

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

## Acknowledgements
Supported by NIH grants DP 1OD579 (G.T.), 1R01MH091326 (G.T.), 1R01MH099231 (G.T. and C.C.), 1P01NS083514 (G.T. and C.C.), Department of Defense W911NF1910280 (C.C. and G.T.), Human Frontier HFSP long-term fellowship LT000009/2017 and Japan-US Brain Research Cooperation Program grant (DM). We thank Robert H. Cudmore (UC Davis) for help with the Map Manager software and Richard L. Huganir and David J. Linden (Johns Hopkins U) for providing SEP-GluA1/myc-GluA1/dsRed2 plasmids and helping with spine imaging.

## Author contributions
D.M. collected the data; D.M. and W.M. analyzed the data; D.M., G.T., and C.C. designed the experiments; D.M., W.M., G.T., and C.C. wrote the paper.

## Competing interests
The authors declare no competing interests.
