## [Peer Review File · Nature Communications]

Reviewer #2 (Remarks to the Author):

The authors have now further added new analysis and carefully revised figures and text addressing most of my previous concerns. I appreciate the redefinition of spines with increasing GluA1 as “max” spines and the added analysis and discussion about how these can reflect learning. I also welcome the replotting of Fig 4a,b and S6, which significantly add clarification to a main point of this study.

I only have one issue remaining regarding the wording of “relative protection” of max spines during sleep. The authors show and state that this protection is provided through a sleep dependent decrease of GluA1 in “other” spines. However, the use of “protection of max spines” suggests a process that is directly affecting these “max” spines. The data provides no evidence for “protection” of max spines but does support the notion of increased signal to noise of max spines to other spines. I would suggest that the authors remove wording regarding “protection” and focus on wording like “sleep may still provide a relative advantage to the max spines by weakening all the remaining spines” (lines 301-302). In particular, the use of “relative protection” in the abstract (lines 42-44) or “lower than expected decrease” in key sections of the results (lines 307-309, 317) might mislead the reader. I would recommend the authors focus the phrasing on the decrease of “other” spines rather than “protection” of “max” spines. This would also be in agreement with the correlation of day 2 performance and changes in GluA1 at 7h, which seems to be mostly driven by the “other” spines.

Provided these wording changes, I believe this important and interesting study is suitable for publishing at Nature Communications.

Reviewer #5 (Remarks to the Author):

Summary

In this study, the authors fluorescently label the GluA1 AMPAR subunit and use in vivo imaging to track how individual synapses change during motor learning and how memory consolidation is impacted by sleep. The authors performed very challenging longitudinal imaging experiments, where they tracked changes in surface GluA1 expression of individual synapses during a motor learning task in the presence or absence of sleep deprivation. They found that motor learning results in synaptic plasticity, observed as an increase in the surface expression of SEP-GluA1 in spines of L1 motor cortex. They further show that this process depends on sleep, as mice that are allowed to sleep following motor learning largely reset these potentiated synaptic networks, with the notable exception of so-called “max” spines that remain potentiated, whereas preventing animals from sleeping after motor learning prevents this reset of the synaptic network.

The fundamental questions explored in this paper are significant and of broad interest to neuroscience and systems biology. The experimental techniques are innovative and appropriate, and the data are rigorously analyzed. This high degree of enthusiasm for the paper is somewhat

diminished by the experimental design, especially the lack of control experiments that would enable the field to directly observe the degree to which synaptic plasticity encodes motor learning vs. changes in sleep state. Though the authors have not performed the key control experiment of imaging mice at different phases of the light/dark cycle with/without sleep deprivation in the absence of motor learning, they have made some efforts to address similar concerns of previous reviewers by imaging mice with minimal motor training. While we believe that including these seemingly crucial control experiments would significantly bolster the strength of the paper, we accept that the current environment given the COVID-19 pandemic may preclude or hinder performing additional longitudinal imaging experiments.

Based on the novelty and significance of the questions being addressed, as well as the high quality of the data and analysis, we feel that this paper merits publication.

Major points

1. If conducting further experiments is possible, the authors should repeat their longitudinal imaging paradigm in one additional control group: mice that receive normal sleep vs. sleep deprivation in the absence of motor learning. This crucial control would enable the authors to directly test how sleep deprivation alone perturbs the natural rhythm of the synaptic network, and will thus enable them to statistically separate the effects of sleep deprivation from motor learning. Without this control, that authors may only investigate how synaptic dynamics are impacted by sleep or deprivation in a group of mice that already received motor training, which they directly demonstrate causes separate but overlapping changes in plasticity. To fully dissect these distinct but interdependent processes, they must be analyzed separately. Without a no motor training control group, this analysis is not possible.
2. The previous point is particularly important for understanding the role of “max” spines. The authors do show the presence of “max” spines in both sleep deprived and non-sleep deprived conditions, suggesting that they are specifically formed by motor learning, but directly demonstrating their absence in mice lacking motor training would definitively prove this. The minimal training condition obliquely addresses this point, but an untrained control group would be ideal.
3. The data are rigorously analyzed throughout the study. All statistical analyses are appropriate. The authors’ use of a linear mixed effect model to compare experimental categorical fixed effects and random control effects to fully explore changes in synaptic weights is particularly strong.
4. I may have missed it, but how long does the motor-learning related synaptic potentiation persist? Did the authors ever look beyond the 24-hr timepoint? In particular, how long was the potentiation observed in max spines?
5. What percent of spines are max spines? If the rest of the synaptic network is almost completely “reset” by a single sleep cycle, then the authors would seem to be suggesting that motor learning is encoded almost entirely by these max spines (at least insofar as synaptic plasticity in L1 motor cortex is encoding learning of this task). It is therefore crucial that the reader understand how prevalent this population of max spines is. If these spines do disproportionately encode learning/memory, this finding will surely spark many future studies to determine their specific molecular characteristics.
6. The observation in Fig. 2 and Table S3 that a net decrease in SEP-GluA1 expression occurs after learning only in animals that are allowed to sleep is interesting. The authors have conclusively shown

that sleep deprivation impairs behavioral performance. So, in the sleep-deprived cohort, the learning-related synaptic potentiation of GluA1 synapses in motor cortex persists despite a decreased ability to perform the task. What does this mean for the mechanism that synaptic plasticity encodes this motor learning? This would seem to merit further discussion, especially since they demonstrate that sleep largely resets the synaptic network.

Minor points

1. The authors are using a dsRed cell fill to image dendritic spines and an overexpressed SEP-GluA1 construct to image AMPARs. They are somewhat inconsistent with their usage of the terms spines (which would presumably be exclusively related to observations of spine size, as measured by dsRed) and synapses (which would presumably be related to observations of either SEP intensity or size). For instance, in many places in the manuscript, the author use the phrases “spines that go up” or “spines up”, though they seem to be describing changes in the SEP signal. The authors should be more clear with their language in describing changes in synaptic plasticity (SEP-AMPArs) and structural plasticity (dsRed spines).
2. This is a completely stylistic point, but in lieu of stating “spines that go up,” the authors may want to consider using a more descriptive phrase such as “synapses that potentiate” or “synapses that strengthen”.
3. One citation seem to be duplicated: Roth et al., 2020a and 2020b are the same paper.

REVIEWER COMMENTS

Reviewer #2 (Remarks to the Author):

The authors have now further added new analysis and carefully revised figures and text addressing most of my previous concerns. I appreciate the redefinition of spines with increasing GluA1 as “max” spines and the added analysis and discussion about how these can reflect learning. I also welcome the replotting of Fig 4a,b and S6, which significantly add clarification to a main point of this study.

I only have one issue remaining regarding the wording of “relative protection” of max spines during sleep. The authors show and state that this protection is provided through a sleep dependent decrease of GluA1 in “other” spines. However, the use of “protection of max spines” suggests a process that is directly affecting these “max” spines. The data provides no evidence for “protection” of max spines but does support the notion of increased signal to noise of max spines to other spines. I would suggest that the authors remove wording regarding “protection” and focus on wording like “sleep may still provide a relative advantage to the max spines by weakening all the remaining spines” (lines 301-302). In particular, the use of “relative protection” in the abstract (lines 42-44) or “lower than expected decrease” in key sections of the results (lines 307-309, 317) might mislead the reader. I would recommend the authors focus the phrasing on the decrease of “other” spines rather than “protection” of “max” spines. This would also be in agreement with the correlation of day 2 performance and changes in GluA1 at 7h, which seems to be mostly driven by the “other” spines.

Provided these wording changes, I believe this important and interesting study is suitable for publishing at Nature Communications.

RE: thank you for the positive and constructive comments. We have reworded the text and substituted “protection” with “advantage”. The sentence (lines 307-309) referring to “lower than expected decrease” has been removed.

Reviewer #5 (Remarks to the Author):

Summary

In this study, the authors fluorescently label the GluA1 AMPAR subunit and use in vivo imaging to track how individual synapses change during motor learning and how memory consolidation is impacted by sleep. The authors performed very challenging longitudinal imaging experiments, where they tracked changes in surface GluA1 expression of individual synapses during a motor learning task in the presence or absence of sleep deprivation. They found that motor learning results in synaptic plasticity, observed as an increase in the surface expression of SEP-GluA1 in spines of L1 motor cortex. They further show that this process depends on sleep, as mice that are allowed to sleep following motor learning largely reset these potentiated synaptic networks, with the notable exception of so-called “max” spines that remain potentiated, whereas preventing animals from sleeping after motor learning prevents this reset of the synaptic network.

The fundamental questions explored in this paper are significant and of broad interest to neuroscience and systems biology. The experimental techniques are innovative and appropriate, and the data are rigorously analyzed. This high degree of enthusiasm for the paper is somewhat diminished by the experimental design, especially the lack of control experiments that would enable the field to directly observe the degree to which synaptic plasticity encodes motor learning vs. changes in sleep state. Though the authors have not performed the key control experiment of imaging mice at different phases of the light/dark cycle with/without sleep deprivation in the absence of motor learning, they have made some efforts to address similar concerns of previous reviewers by imaging mice with minimal motor training. While we believe that including these seemingly crucial control experiments would significantly bolster the strength of the paper, we accept that the current environment given the COVID-19 pandemic may preclude or hinder performing additional longitudinal imaging experiments.

Based on the novelty and significance of the questions being addressed, as well as the high quality of the data and analysis, we feel that this paper merits publication.

Major points

1. If conducting further experiments is possible, the authors should repeat their longitudinal imaging paradigm in one additional control group: mice that receive normal sleep vs. sleep deprivation in the absence of motor learning. This crucial control would enable the authors to directly test how sleep deprivation alone perturbs the natural rhythm of the synaptic network, and will thus enable them to statistically separate the effects of sleep deprivation from motor learning. Without this control, that authors may only investigate how synaptic dynamics are impacted by sleep or deprivation in a group of mice that already received motor training, which they directly demonstrate causes separate but overlapping changes in plasticity. To fully dissect these distinct but interdependent processes, they must be analyzed separately. Without a no motor training control group, this analysis is not possible.

RE: We very much appreciate the Reviewer's acknowledgement that SEP-GluA1 repeated in vivo imaging is difficult. This is even more so during the persisting challenges caused by the pandemic, which indeed preclude us from running additional control experiments. We wish to mention, in addition, two key points:

- 1- the specific question raised by the Reviewer – how sleep deprivation alone, without training in a specific task, affects the synaptic network – is important and is the first question that we addressed in previous studies. In one study (Vyazovskiy et al Nature Neurosci 2008) we compared the effects of sleep, spontaneous wake and sleep deprivation on GluR1 expression and GluR1-Ser845 expression in synaptoneurosome from the rat cortex, as well as on electrophysiological measures of synaptic strength. In another study (De Vivo et al. Science 2017) we used serial electron microscopy to measure the effects of sleep, spontaneous wake and sleep deprivation on an ultrastructural measure of synaptic strength, the axon-spine interface, in the spines of mouse cortex. These studies did not use SEP-GluA1 repeated in vivo imaging but relied on different and complementary methods. They reached the same general conclusion

when comparing sleep with wake conditions without training in a specific task: various measures of synaptic strength are lower after sleep than after wake, with no major differences between spontaneous wake and sleep deprivation.

- 2- The current study was designed as the next step, to address two different and specific questions still standing in the sleep field: 1) whether the net downregulation of synaptic strength afforded by pre-learning sleep still happens when sleep follows training (see also point 6); 2) whether the synapses affected by training are “treated” by sleep differently than the other synapses, and specifically, as many believe, whether the synapses strengthened by learning can be further potentiated by sleep. Of course, adding additional controls would have provided more context to interpret the current findings. However, the lack of these controls does not directly affect the conclusions we could draw with respect to these two key questions.

2. The previous point is particularly important for understanding the role of “max” spines. The authors do show the presence of “max” spines in both sleep deprived and non-sleep deprived conditions, suggesting that they are specifically formed by motor learning, but directly demonstrating their absence in mice lacking motor training would definitively prove this. The minimal training condition obliquely addresses this point, but an untrained control group would be ideal.

RE: We agree: our minimal training condition suggests, but does not prove, that max spines are formed due to motor learning. We stated before that we assume based on our analyses that there is a link between motor learning and, at least, a subset of max spine. We now state explicitly that more direct evidence would be needed to prove that max spines are directly related to learning. Note, however, that it is unlikely that max spines will be totally absent in mice not trained, because we have seen that a few spines show large SEP-GluA1 changes at all time points, with or without training, or sleep, or wake, consistent with what was reported in other experimental conditions (e.g. Roth et al., 2020).

3. The data are rigorously analyzed throughout the study. All statistical analyses are appropriate. The authors’ use of a linear mixed effect model to compare experimental categorical fixed effects and random control effects to fully explore changes in synaptic weights is particularly strong.

RE: Thank you for this comment.

4. I may have missed it, but how long does the motor-learning related synaptic potentiation persist? Did the authors ever look beyond the 24-hr timepoint? In particular, how long was the potentiation observed in max spines?

RE: Our latest imaging timepoint was 24 hours after the first training. We now state that we do not know for how long the potentiation lasts beyond this timepoint. We also cite a recent mouse study in which after 8 days of training in the reaching task, 8.3% of all spines showed an increase in SEP-GluA1 of at least 26%, and this subset of spines remained potentiated a week after the last day of training (Roth et al Neuron 2020).

5. What percent of spines are max spines? If the rest of the synaptic network is almost completely “reset” by a single sleep cycle, then the authors would seem to be suggesting that motor learning is encoded almost entirely by these max spines (at least insofar as synaptic plasticity in L1 motor cortex is encoding learning of this task). It is therefore crucial that the reader understand how prevalent this population of max spines is. If these spines do disproportionately encode learning/memory, this finding will surely spark many future studies to determine their specific molecular characteristics.

RE: Max spines represent $13.5 \pm 4.1\%$ of all spines (207/1530; mean \pm sd; range per mouse = 6.3% - 20%), as now stated in the main text and in the legend of figures 4 and S6. As mentioned above, Roth and colleagues (Neuron 2020) found that 8.3% of all spines showed an increase in SEP-GluA1 of at least 26% in the last 2 days of training compared to baseline (147/1781, M1 apical dendrites of layer 5 pyramidal neurons).

6. The observation in Fig. 2 and Table S3 that a net decrease in SEP-GluA1 expression occurs after learning only in animals that are allowed to sleep is interesting. The authors have conclusively shown that sleep deprivation impairs behavioral performance. So, in the sleep-deprived cohort, the learning-related synaptic potentiation of GluA1 synapses in motor cortex persists despite a decreased ability to perform the task. What does this mean for the mechanism that synaptic plasticity encodes this motor learning? This would seem to merit further discussion, especially since they demonstrate that sleep largely resets the synaptic network.

RE: Thank you for raising this important point. The finding that the overall effect of sleep remains a net decrease in synaptic strength even after learning, and even when focusing on a cortical region that is directly involved in the task, is indeed interesting. In fact, it was one of the key questions that this study was set to address, because many of our sleep colleagues assumed that the net, sleep-dependent, synaptic weakening would not occur in the circuit directly engaged in skill learning (see discussion, lines 413-423).

At the beginning of session 2 the SD mice are worse than the S mice, but they nonetheless are able to perform the task well. Our results suggest that the GluA1 changes in max spines, or at least in some of the max spines, are the likely candidate to account for the successful skill learning in all mice, whether or not they were allowed to sleep. Sleep allows a small but significant additional improvement in performance in this task, consistent with the extensive literature on sleep and memory consolidation in both animals and humans. Our results show that this advantage seems to come exclusively from reducing the noise (other spines), not by increasing the signal (max). This is an intriguing result, and we now discuss in more detail the implications of these findings as follows:

“At the beginning of the second training session all mice remembered well how to perform the task. However, mice allowed to sleep performed significantly better than sleep deprived mice, consistent with previous findings in rodents and humans⁵. Max spines are likely to reflect successful skill learning, and indeed they remained strong post-learning in both S and SD mice. On the other hand, max spines did not differ between the two groups, and their level was not correlated with performance at the beginning of the second training session, suggesting that they

are not directly involved in the performance advantage afforded by post-learning sleep. Conversely, good performance in the second session correlated with the downregulation of SEP-GluA1 after the first learning session in all spines or in the other spines, which were the great majority of spines. Together, these findings suggest that sleep benefits memory consolidation by increasing the signal to noise ratio, and does so by decreasing the noise (other spines) rather than by increasing the signal per se (max spines). In other words, the advantage provided by sleep seems to depend on its weakening effect on the other spines, rather than on a direct effect on the max spines. In a rat model of neuroprosthetic learning, the firing of a small group of task-related neurons whose activity was causally linked to learning the task increased slightly after post-learning sleep, whereas firing decreased substantially in a larger set of task-unrelated neurons^{47,48}. Future experiments in which both neuronal and synaptic activity are measured in the same animal after learning and post-learning sleep could in principle determine whether the weakening effect of sleep on the other spines can account for the effects on neuronal firing.”

Minor points

1. The authors are using a dsRed cell fill to image dendritic spines and an overexpressed SEP-GluA1 construct to image AMPARs. They are somewhat inconsistent with their usage of the terms spines (which would presumably be exclusively related to observations of spine size, as measured by dsRed) and synapses (which would presumably be related to observations of either SEP intensity or size). For instance, in many places in the manuscript, the author use the phrases “spines that go up” or “spines up”, though they seem to be describing changes in the SEP signal. The authors should be more clear with their language in describing changes in synaptic plasticity (SEP-AMPA) and structural plasticity (dsRed spines).

RE: Thank you for prompting us to be more consistent. Since we cannot rule out that at least a small portion of the SEP-GluA1 signal in the spines is extrasynaptic, we now refer to this signal as spine SEP-GluA1 signal or, spine-surface GluA1 expression.

2. This is a completely stylistic point, but in lieu of stating “spines that go up,” the authors may want to consider using a more descriptive phrase such as “synapses that potentiate” or “synapses that strengthen”.

RE: Up is used for short to refer to the positive normalized difference in SEP-GluA1 expression. Although not ideal, based on previous comments, we prefer to keep this nomenclature because “strengthened/potentiated” would imply, at least in the mind of some readers, that we have direct proof that the synaptic strength of the Up spines is increased (e.g. increased AMPA currents).

3. One citation seem to be duplicated: Roth et al., 2020a and 2020b are the same paper.

RE: Fixed.